# Lunasin as a Promising Plant-Derived Peptide for Cancer Therapy

**DOI:** 10.3390/ijms23179548

**Published:** 2022-08-23

**Authors:** Stephanny Miranda Alves de Souza, Blanca Hernández-Ledesma, Theo Luiz Ferraz de Souza

**Affiliations:** 1Faculdade de Farmácia, Universidade Federal do Rio de Janeiro, Rio de Janeiro 21941-902, Brazil; 2Programa de Pós-Graduação em Nanobiossistemas, Duque de Caxias 25240-005, Brazil; 3Instituto de Investigación en Ciencias de la Alimentación (CIAL, CSIC-UAM, CEI UAM + CSIC), C/Nicolás Cabrera 9, 28049 Madrid, Spain

**Keywords:** lunasin, therapeutic peptides, cancer therapy, anticancer mechanism of action, structural features, pharmacokinetics

## Abstract

Cancer has become one of the main public health problems worldwide, demanding the development of new therapeutic agents that can help reduce mortality. Lunasin is a soybean peptide that has emerged as an attractive option because its preventive and therapeutic actions against cancer. In this review, we evaluated available research on lunasin’s structure and mechanism of action, which should be useful for the development of lunasin-based therapeutic products. We described data on its primary, secondary, tertiary, and possible quaternary structure, susceptibility to post-translational modifications, and structural stability. These characteristics are important for understanding drug activity and characterizing lunasin products. We also provided an overview of research on lunasin pharmacokinetics and safety. Studies examining lunasin’s mechanisms of action against cancer were reviewed, highlighting reported activities, and known molecular partners. Finally, we briefly discussed commercially available lunasin products and potential combination therapeutics.

## 1. Introduction

Cancer is one of the leading causes of death worldwide. The International Agency for Research on Cancer (IARC) reported 19.3 million cases and 10 million deaths occurring in 2020 [1]. Case numbers are expected to increase significantly over the next few years. By 2040, cancer cases are projected to reach 28.4 million, corresponding to a 47% increase from 2020 [1]. These data reveal that cancer has become a major global public health crisis, demanding increasingly efficient therapies to reduce mortality. Conventional therapies for cancer treatment include surgery, radiotherapy, chemotherapy, immunotherapy, and hormonal therapy, as well as their combinations [2]. Each method has advantages and limitations. Some difficulties encountered in cancer treatment are related to targeting cancer stem cells, drug resistance, side effects, and metastasis [3]. New therapeutic agents that can solve one or more of these problems are essential for increasing treatment efficiency.

Plant-derived peptides are promising antitumor agents with various mechanisms of action [4]. Lunasin is a 43 amino-acid soybean-derived peptide with demonstrated anticancer activity [5,6]. Early in vitro and in vivo studies on lunasin indicated that its preventive action occurred through inhibiting transformation events. Subsequent studies have demonstrated that lunasin acts against established cancer cell lines and xenograft tumors, strengthening the idea of its therapeutic application. Given these anticancer effects, lunasin has the potential for development into both pharmaceutical and nutraceutical products. The antioxidant, anti-inflammatory, and immunomodulatory activities in lunasin may enhance its action against cancer. In addition, lunasin can act on cancer stem cells that are associated with problems encountered during cancer treatment, such as recurrence, drug resistance, and metastasis development [7]. Lunasin also has an epigenetic mechanism of action that is related to histone acetylation. Overall, lunasin is a promising candidate for therapeutic application.

In this review, we aimed to provide essential information for the development of lunasin as a therapeutic peptide against cancer. Here, we examine the structure, pharmacokinetics, safety, and mechanisms of action from a pharmaceutical perspective. We also discuss possible therapeutic applications and their challenges.

## 2. Lunasin Structure

Structural information of lunasin needs to be clearly understood before its implementation in therapy. Therapeutic peptides do not possess specific definitions, and regulatory guidelines differ depending on their size and manufacturing procedure [8,9,10]. Lunasin can be obtained in different ways, including chemical synthesis, biotechnology, and extraction from natural soybean sources. In general, structural data is important for evaluating the efficacy, safety, and quality of pharmaceutical products. For biological and biotechnological products, guidelines such as ICH Q5E, Q5C, and Q6B require determining the physicochemical properties of primary, secondary, tertiary, and quaternary structures, structural heterogeneity derived from post-translational modifications (PTMs), and stability. These traits are important for quality control and the evaluation of changes during manufacturing to ensure comparability.

### 2.1. Structure-Activity Relationship

Lunasin is divided into four regions: (1) N-terminus, comprising the first 22 residues; (2) the central portion, presenting similarity with chromodomains of chromatin-binding proteins; (3) the RGD motif; and (4) C-terminus, rich in aspartic acid (D) [11,12]. These different parts of the sequence are differentially correlated with actions already reported for this peptide (Table 1). The full sequence is important to exert cytotoxicity to MDA-MB-231 breast cancer cell line [13], anti-transformation [14], and anti-inflammatory [15] activities.

The most notable part of the lunasin sequence is the aspartic acid (D)-rich tail. Because this portion is associated with the capacity of disrupt histone acetylation, it is considered fundamental for lunasin’s anticancer properties [5,13]. Lunasin binding to deacetylated histones occurs via electrostatic interactions between the aspartic (D)-acid tail and basic residues from the N-terminus of histones H3 and H4 [5]. Data suggest that the aspartic acid (D)-tail is primarily responsible for the lunasin-mediated inhibition of histone H3 acetylation, whereas other parts of its sequence are also required for histones H4 [13,14]. Furthermore, the aspartic acid (D)-tail plays a role in antimitotic effects through disrupting spindle fiber formation [5].

The RGD cell adhesion motif recognizes integrins (especially αvβ3, αvβ5, and α5β1) involved in cancer progression, metastasis, and several related processes, including angiogenesis, cell proliferation, differentiation, and apoptosis [19,20]. This motif is important for lunasin cell adhesion but apparently unnecessary for binding with histones [14]. The RGD motif is also important for lunasin internalization into cells and its anti-transformation effect [14]. However, because the RGD motif is not necessary for lunasin internalization into NIH3T3 cells, its involvement in internalization seems cell-specific [16]. The RGD motif has been linked to the lunasin-induced inhibition of oncosphere formation in melanoma cell line A375 ALDH^high^ [21].

The lunasin sequence has a similar central portion as chromatin-binding proteins; this section is thus related to lunasin’s capacity for chromatin binding and its anti-transformation effect [14,16]. Moreover, the central region causes cytotoxicity in the breast cancer cell line MDA-MB-231 [13].

The function of the lunasin N-terminus is unknown [11,12]. However, some studies have implicated the N-terminus in known effects reported for lunasin [13,14,15,16,17,18]. This region increases binding affinity for the inhibition of histone H4 acetylation [14]. It is also important for the anti-transformation effect and exhibits some cytotoxicity against cancer cells [13,16]. The N-terminus and central regions of the lunasin sequence are related to other actions that could enhance anticancer activity and provide other health benefits. Overall, both regions are considered responsible for lunasin’s antioxidant and immunomodulatory activities [15,17,18].

The available data indicate that every part of the lunasin sequence makes vital contributions to different mechanisms of action. Thus, any proposed modification to its primary structure when developing a lunasin therapeutic product should be assessed regarding potential negative effects on total activity.

### 2.2. Physicochemical Characterization

#### 2.2.1. Primary Structure and Post-Translational Modifications

Lunasin was first isolated and sequenced in 1987 as a 43 amino acid peptide with the sequence SKWQHQQDSCRKQLQGVNLTPCEKHIMEKIQGRGDDDDDDDDD [6]. Some researchers have also reported soybean-derived lunasin with an additional asparagine (N) residue at the C-terminus [22,23]. Thus, naturally occurring lunasin can be either 43 or 44 amino acids in length. A feature of its primary structure is the high presence of acid residues, particularly aspartic acid (D), accounting for almost a quarter of the total sequence. As a result, lunasin has a theoretical pI of approximately 4.43 (https://web.expasy.org/protparam/, accessed on 8 June 2021). Its sequence also contains many hydrophilic and charged residues that make the peptide intrinsically disordered [24].

Clarifying the sequence would help to reveal the possible sites of protease cleavage. Proteolytic degradation is a major weakness of therapeutic peptides that limits their systemic applications [25]. Although lunasin is bioavailable after oral ingestion in both humans and animal models, it undergoes intense protease digestion in the gastrointestinal tract [26,27,28,29]. At least 12 cleavage sites have been suggested for digestive (pepsin, trypsin, and chymotrypsin) and plasmatic (thrombin) proteases using PeptideCutter from Expasy (https://web.expasy.org/peptide_cutter/, accessed on 8 June 2021) (Figure 1). More data on cleavage sites will be useful for optimizing lunasin bioavailability.

PTMs can occur during manufacture and storage, inducing structural heterogeneity that affects product quality (ICH Q6B). Identifying these alternate forms is important, because regulatory bodies consider them product-related substances when the compounds have activity, efficacy, and safety comparable to the main product; when they do not, the modified substances are considered impurities (ICH Q6B). Since lunasin can be consumed through soybean-derived foods, it is important to know the PTMs formed by food processing. A study that investigated lunasin isolated from commercially processed foods found that it contains diverse PTMs, most notably the glycation of K_24_ and K_29_ [23]. These two sites are present in the region that shows similarity with the chromodomain of chromatin-binding proteins. Furthermore, this study identified six lunasin-derived advanced glycation end products (AGEs) that can be generated in vivo and in food processing during the oxidation of Maillard reaction derivatives [23]. Because AGEs contribute to some human disorders [30], lunasin-derived AGEs can lower the safety of any proposed lunasin nutraceutical and functional food products, and, therefore, require additional investigation.

Other side modifications identified in processed-food-derived lunasin include oxidation (M), dihydroxy (K), dehydration (D), deamidation (N/Q), methyl esterification (D), carbamylation (K), acetylation (K), and pyroglutamate conversion (N-terminal Q) [23]. Most of these PTMs are non-enzymatic (spontaneous chemical reactions), while others are enzymatically produced. Even non-enzymatic PTMs could occur naturally in plants, such as disulfide bond formation, oxidation, dihydroxylation, deamidation, and carbamylation [31]. However, some identified PTMs, such as oxidation, deamidation, and pyroglutamate, may be induced by stress conditions during food processing or be artifacts produced in sample preparation to analysis [31]. Pyroglutamate formation occur non-enzymatically or enzymatically in glutamine or glutamate residues located at the N-terminus [32] and could only be observed in specific cleaved forms of lunasin. Although these modifications have been detected in lunasin present in processed foods, they may not occur naturally or be relevant in the soybean-extracted, recombinant, and synthetic peptide.

The most significative PTM reported in lunasin sequence is the formation of disulfide bond. Disulfide bonds are important PTMs that affect protein folding and stability [33]. For example, the formation of intra- and intermolecular disulfide bonds can stabilize tertiary and quaternary structures [34]. Structural changes and aberrant disulfide bond formation may occur during the manufacturing of therapeutic proteins, presenting a challenge [35]. The primary structure of lunasin contains two cysteine (C) residues (at positions 10 and 22 of the polypeptide chain) that can form intramolecular disulfide bond as has been reported in synthetic and recombinant forms [24,36]. Cysteines on the lunasin sequence are either reduced or oxidized, depending on environmental conditions [36]. In aqueous solutions, lunasin cysteines tend to be oxidized, eventually forming intramolecular bonds [24,36]. The biological implications of this intramolecular bond formation remain unclear but may be related to lunasin’s anti-inflammatory and antioxidant activities [36].

#### 2.2.2. Secondary Structure

Both in silico and in vitro approaches have yielded data on the secondary structure of lunasin. Initial studies recognized a region possessing structural homology with chromatin-binding proteins and suggested that it was an α-helix motif [14]. Molecular dynamic (MD) simulations with an extended structure of the 43-residue lunasin sequence suggested considerable structural content, with three separate α-helices in the N-terminal (H_5_-C_10_), central (C_33_-I_30_), and C-terminal (D_35_-D_41_) regions [37]. Dia et al. analyzed the secondary structure of lunasin purified from soybean using circular dichroism in PBS (pH 7.5). At 25 °C, spectrum deconvolution suggested a structural content of 29% α-helix, 28% β-strands, 23% turns, and 20% unordered [38].

Studies performed with recombinant and synthetic lunasin have indicated a mostly unordered structure with transient α-helices [24,36]. Transient helices can be favored and stabilized depending on the conditions or on specific binding to certain molecular partners. Protonation of aspartic acid (D) residues did not reveal significant alterations in the secondary structure [24,36]. Available data suggest that lunasin is an intrinsically disordered peptide in a pre-molten globule-like state [24]. Structural NMR studies demonstrated that lunasin is unordered, with two α-helices and a β-strand at the C-terminus as transient secondary-structural elements. Additionally, lunasin secondary structure was similar with or without intramolecular disulfide bond formation [36].

Figure 2 compiles data from different analyses that indicate the location of the secondary structure content in the lunasin sequence. Most of these data are derived from in silico analysis, and only the NMR describes the secondary structure content observed in vitro. Different conditions, such as pH, the formation of the disulfide bond, and presence of 2,2,2-trifluoroethanol (TFE), are considered. The α-helix motif with similarity to chromatin-binding proteins appears to be a common element in these data (Figure 2). As observed by the NMR data, lunasin has two regions prone to adopting the transient α-helix content.

#### 2.2.3. Tertiary and Quaternary Structures

The most striking feature of the lunasin tertiary structure is its intrinsic disorder [24,36]. This means that it has no well-defined three-dimensional structure in solution, possessing flexibility that permits multiple conformations [39,40]. Plasticity provides the capacity to bind with different partners through structural modification, possibly explaining its multiple effects [24,40]. Lunasin structural plasticity must be considered during quality control, given its important biological implications for understanding efficacy and safety. Another remarkable feature of lunasin tertiary structure is compaction [24]. Experimental determination of the lunasin Stokes radius indicates a more compact structure than expected for a disordered peptide, which is supported by computational analyses [24]. Lunasin achieves its compactness through intramolecular disulfide bonds and electrostatic interactions between charged residues [24]. The disulfide bond also affects structural flexibility, with the oxidized form being less flexible and more stable than the reduced form [24].

Several studies have identified lunasin as both monomeric and dimeric peptide [28,41]. Thus, a study observed the monomeric form in rat blood after feeding subjects lunasin-enriched soy, also finding the dimeric form in rat livers [28]. The dimeric form was also identified in the livers of rats fed lunasin-enriched wheat [41]. Thus, although it is monomeric in solution, lunasin dimerization may be favored under some conditions and cannot be ruled out.

#### 2.2.4. Stability

The thermal stability of lunasin has been investigated in several studies [24,38]. One study used circular dichroism spectroscopy (CD) to assess the secondary structure and thermal stability of lunasin purified from defatted soybean flour. The results showed that lunasin was stable up to 72 °C temperature but denatured by 90 °C [38]. Purified lunasin heated to 100 °C exhibited a 24.4% decrease in binding affinity with a rabbit polyclonal antibody, suggesting that this temperature caused a structural change [38]. A thermal stability evaluation of synthetic lunasin further indicated stability up to 100 °C [24]. In contrast, the analysis of CD spectra at different temperatures found heating to increase secondary structures slightly, a characteristic of intrinsic disorder that favored hydrophobic interactions. Heated synthetic lunasin also showed a slight difference in structural content from non-heated lunasin at both pH 1.5 and 7.4 [24]. Additionally, heated synthetic lunasin did not exhibit aggregation according to CD analyses. Taken together, these data suggest that lunasin is heat-stable, valuable knowledge for the processing and storage of lunasin products.

## 3. Pharmacokinetics and Safety

Regulatory guidelines indicate that the pharmacokinetic (PK) documentation of therapeutic polypeptides contribute to ensuring efficacy and safety in patients [42]. PK involves determining the absorption, distribution, metabolism, and excretion (ADME) of the tested drug [43]. Larger molecules differ from small molecules in ADME because of higher molecular weight and structural complexity [44].

Therapeutic peptides are not usually administered orally because they have low systemic bioavailability, given high gastrointestinal enzyme activity and modest permeation through gastrointestinal mucosa [45]. In vitro studies have shown that lunasin is highly digested by digestive enzymes, although the percentage of residual lunasin differ among studies [27,29,46,47]. Assay conditions, detection method, and lunasin source (synthetic, purified from soybeans, or lunasin-enriched soybean products) are all factors that could influence these results. Nevertheless, the experiments revealed that lunasin must be protected from digestion if it is orally administered. Protease inhibitors in soybean, such as the Bowman–Birk inhibitor and Kunitz trypsin inhibitor, may protect lunasin from digestion, enabling bioavailability after oral intake [27,28,29,48]. In addition to the co-administration with protease inhibitors, other strategies can be employed to allow oral administration of lunasin. Some strategies to enhance oral bioavailability and increase resistance to proteolytic digestion may include chemical modifications (e.g., D-amino acids), mucoadhesive systems (e.g., chitosan, carbopol, cellulose derivatives) and carrier systems (e.g., liposomes, hydrogels, polymeric micro- and nanoparticles) [49,50].

Absorption across the intestinal mucosa is crucial for oral bioavailability. Lunasin is absorbed through the Caco-2 cell line monolayer, with paracellular passive diffusion as the main route of transport [51]. The investigation of lunasin bioavailability in the plasma of humans on a soybean-based diet revealed an average of 4.5% lunasin, indicating that the peptide resisted gastrointestinal digestion and was absorbed in vivo [26]. Oral bioavailability was also investigated in animals. Approximately 30% of orally administered lunasin was absorbed in mice that received a mixture of ^3^H-labelled synthetic lunasin and lunasin-enriched soybean (LES) [28]. In both human and mouse studies, protease inhibitors help achieve lunasin bioavailability. Researchers have also investigated whether digestion-resistant lunasin and its derived peptides could remain bioactive. Lunasin extracted from the liver of rats fed LES successfully suppress foci formation, similar to synthetic lunasin [28]. Additionally, lunasin extracted from the liver of rats fed with lunasin-enriched wheat still had the ability to inhibit histone H3 and H4 acetylation [41]. After lunasin is digested in vitro by simulated gastrointestinal fluids, it is still internalized into cells and localized in the nucleus [27]. Finally, lunasin subjected to a simulated digestion assay with pepsin and pancreatin exerted an antiproliferative effect against colon cancer cell lines [29].

Researchers have also investigated how lunasin is absorbed through intranasal and topical administration [14,52,53,54]. Intranasally administered lunasin reached the central nervous system and exerted an antipsychotic effect [52], demonstrating that the intranasal route has therapeutic potential. The intranasal application of lunasin was also evaluated for the ability to treat asthma, with the results indicating the alleviation of airway inflammation and again demonstrating the efficacy of this route [54]. When administered topically, lunasin can penetrate the skin and localize into cell nuclei. Topical administration decreased skin tumor incidence, shrank the tumors, and delayed their appearance [14]. The topical application of lunasin in mice revealed its antiproliferative effect on keratinocytes [53]. Together, these data provide evidence that the body can absorb lunasin through other routes besides oral administration.

After absorption and reaching systemic circulation, lunasin is distributed throughout the body and localized in various tissues [28]. The oral administration of ^3^H-labelled synthetic lunasin in mice allowed for the detection of lunasin in cancer-prone tissues, such as the lung, mammary gland, and prostate [28]. Lunasin has also been detected in the esophagus, stomach, cecum, colon, liver, kidney, heart, and skin [28,47]. Lunasin can cross the blood–brain barrier and act on the central nervous system [28,52]. To take effect, however, proteases on the cell surface or secreted by cells must be avoided [55]. A study examining lunasin resistance to brush-border peptidases in Caco-2 cells found the peptide to decrease by 12% but did not detect any lunasin fragments, indicating that the decrease could be due to lunasin’s internalization into cells [56]. Another study found that the central region of lunasin was susceptible to cleavage by brush-border peptidases on the apical side of Caco-2 monolayers [51]. Moreover, after 20 h of incubating lunasin with HepG2 cells, less than 1% remained in the medium and lunasin-derived fragments were detected [57]. Thus, depending on the cell type, lunasin may be more susceptible to proteolysis, and their bioactivity would mainly occur through derived fragments.

Several of lunasin’s actions can only take effect if it is internalized into cells. Fortunately, this internalization occurs despite lunasin being a large peptide with many hydrophilic and charged residues. Although a common hypothesis is that the integrin-binding RGD motif is responsible for cellular internalization, peptides without this motif have also been internalized, suggesting that this process is cell-specific [14,16]. In macrophages, lunasin internalization is proposed to occur via αvβ3 integrin signaling and clathrin-dependent endocytosis [58,59].

The elimination of a drug from the body depends on its metabolism and excretion. For therapeutic polypeptides, metabolism occurs mainly through proteolysis, and the resultant amino acids can be utilized for de novo biosynthesis pathways [45,60]. Mediated by peptidases and proteases, this process is not site-specific and can occur throughout the body [43,45], including common metabolic sites (e.g., liver, kidneys, gastrointestinal tract, blood circulation), other organs and tissues, as well as inside cells [43,44,45]. As mentioned earlier, lunasin is highly susceptible to digestion by proteases, thus susceptibility to proteolysis should be investigated for all lunasin therapeutic products to ensure that efficacy is not compromised. Peptides and proteins with low molecular weight (<30 KDa) may undergo renal excretion due to glomerular filtration, although this is also dependent on their structure and net charge [43,44,45]. Its physicochemical properties make lunasin prone to renal elimination, and experiments have already detected lunasin in mouse urine [28].

More studies are required to ensure that lunasin is safe for therapeutic application. To date, some animal studies have suggested that lunasin selectively acts on cancer cells and lacks side effects during in vivo administration. Table 2 summarizes the available data on lunasin activity in normal and immortalized cells. Some studies have suggested that lunasin does not affect the viability of murine fibroblast NIH/3T3 cells at concentrations up to 10 µM [16,61,62]. Assays using the human lens epithelial cell line SRA 01/04 indicated no effect of recombinant lunasin on viability and apoptosis at concentrations up to 100 µM [63,64]. Recombinant lunasin also had no effect on the viability of the human umbilical vein cell line EA.hy926 at concentrations up to 120 µM [65]. While lunasin caused cytotoxicity in colon cancer cell lines KM12L4, HT-29, HCT-116, and RKO, it did not do so in normal human colon fibroblasts CCD-33Co [66]. Lunasin treatment exerted cytotoxic effects on MCF-7 and MDA-MB-231 breast cancer cell lines with increasing concentration and incubation time but did not affect the viability of non-tumorigenic breast cell line MCF-10A [67]. Lunasin did not alter the viability of bronchial epithelial cell lines BEAS-2B and HBE135-E6E7, although it exerted antiproliferative effects against the NCI-H661 lung cancer cell line under anchorage-dependent conditions [68]. Lunasin had no effect on the viability of adipocyte-differentiated 3T3-L1 cells [69] nor on the macrophage cell line RAW 264.7 [15,17,70,71].

Some studies have investigated the effect of lunasin on non-cancer cells proliferation. The lunasin treatment of chondrocytes obtained from normal knee joint cartilage resulted in a concentration-dependent effect on proliferation. Lunasin did not affect cell proliferation up to a concentration of 100 µM; however, treatment with 500 µM decreased proliferation [72]. Lunasin decreased proliferation and induced apoptosis in synovial fibroblasts from patients with rheumatoid arthritis [73]. Notably, this disease has cancer-like pathogenesis involving an inflammatory process. Lunasin also decreased the proliferation of the embryonic kidney 293 cell line (HEK-293) at a concentration of 100 µM [74]. However, this result does not necessarily indicate cytotoxicity to normal kidney cells, as HEK-293 is an immortalized cell line with tumorigenicity and no tissue-specific gene expression [75,76]. Overall, available evidence is insufficient for concluding cytotoxicity to non-cancerous cells.

The administration of lunasin to mice showed no side effects and no change in body weight [28] nor any notable toxicological effects [77]. The latter study found that lunasin-treated mice did not differ in blood cell count or liver enzyme and creatinine levels from control. Additionally, even when lunasin crosses the blood–brain barrier and acts on the central nervous system, it does not influence mouse activity [52]. Nevertheless, more safety assessments are needed given the possibility of affecting the central nervous system.

## 4. Mechanisms of Action against Cancer

Cancer is a complex disease involving genetic and epigenetic alterations [79]. Genetic, epigenetic, and phenotypic changes cause inter- and intratumoral heterogeneity; when this dynamic variation occurs in a single tumor, temporal heterogeneity is also observed [80]. Cancer heterogeneity can lead to drug resistance, limiting treatment effectiveness, and combining therapies has been applied to address this problem [80,81,82]. Therapeutic agents with multiple types of anticancer activity are therefore promising candidates for improving therapeutic effectiveness. Lunasin can act against different cancer cell lines (Table 3), including colon, breast, stomach, lung, melanoma, and leukemia. The effects in these cell lines have varied, with IC_50_ values ranging from 13 to 508 µM. Effectiveness also changed depending on lunasin source in some cell lines, possibly attributable to impurities and structural differences [83]. These variations illustrate the importance of choosing appropriate lunasin sources and purification processes when characterizing a possible pharmaceutical product.

Lunasin was not cytotoxic to some cancer cell lines. Caco-2 cells are derived from human colon carcinoma and spontaneously differentiate into monolayer cells with absorptive enterocyte properties [89]. A study found that lunasin did not alter the viability of Caco-2 cells [56], while another reported a modest decrease in cell viability after lunasin treatment for 24 h and 48 h [51]. Lunasin did not alter cell viability nor induce apoptosis in HepG2 cells [57,63,64], a non-tumorigenic human hepatoma cell line with differentiated hepatic functions that is widely used in pharmaco-toxicological studies [90]. Furthermore, no cytotoxicity was observed in THP-1 cells treated with lunasin at concentrations up to 100 µM [59]. THP-1 is a human leukemic cell line that differentiates into macrophage-like cells with close resemblance to human monocytes [91]. Finally, the treatment of the murine breast cancer cell line 4T1 with up to 50 µM lunasin did not decrease cell viability but inhibited metastasis [69].

### 4.1. Molecular Partners

To act on the central nervous system, lunasin has a modest affinity for dopamine D1 and a low affinity for serotonin 5-HT2A and 5-HT2C receptors [52,92]. Similar to other intrinsically disordered proteins, lunasin’s intrinsically disordered structure and structural plasticity likely favor binding to multiple partners. Thus, we may discover new molecular partners for lunasin in the future.

Histones were the first main molecular partner identified for lunasin. Solid-phase immunoblotting assays revealed that lunasin binds to histones through interaction with the aspartic acid (D)-tail [5]. Lunasin’s histone-binding ability was also demonstrated through immunoblotting with deacetylated H4 NH_2_-terminus [14], immunoprecipitation with recombinant H4 [93], and proximity ligation assays (PLA) with H3 and H4 performed in situ in H661 and H1299 cells [94]. Galvez and de Lumen (1999) proposed that lunasin binds through electrostatic interactions between its aspartic acid (D)-tail and the basic N-terminus of deacetylated histones, such as those encountered in centromeres; this binding then displaces kinetochore proteins and disturbs spindle fiber attachment. Such a mechanism explains the antimitotic effect observed when transfecting the lunasin gene into mammalian cells [5].

Histone binding is related to lunasin’s epigenetic mechanism of action. Numerous studies have demonstrated that lunasin prevents H3 and H4 histone acetylation through histone acetyl transferases (HATs) [13,14,21,47,94,95,96]. Therefore, lunasin competes with HATs as a ligand for deacetylated histones. Histone acetylation/deacetylation affects chromatin structure and controls gene expression. Acetylation is generally involved in transcription activation, while deacetylation is involved in suppressing gene expression [97]. Maintaining deacetylated histones, lunasin represses gene expression. Lunasin can also promote the acetylation of specific lysine (K) residues while deacetylating others, resulting in the upregulation of genes [93]. Thus, its histone binding appears to regulate gene expression. In addition to the aspartic acid (D)-tail, the N-terminus and central regions of lunasin are important for affinity to the deacetylated N-terminus of H4 [14].

Integrins were the second molecular partner to be identified for lunasin. Co-immunoprecipitation experiments demonstrated interactions with α5β1 [98], co-immunoprecipitation and liquid chromatography coupled to tandem mass spectrometry (LC-MS/MS) identified αvβ3 interactions [58], while PLA identified interactions with the αv, α5, β1, and β3 subunits [94], as well as with the αv subunit [21]. RGD is a well-known cell-adhesion motif recognized by eight integrin subtypes (αvβ1, αvβ3, αvβ5, αvβ6, αvβ8, α5β1, α8β1, and αIIbβ3). Subunit selectivity is related to conformation specificity [20,99,100]. Integrins are transmembrane heterodimeric cell surface receptors comprising α and β subunits, the primary function of which is transmitting signals into cells that determine migration, survival, and differentiation [101]. Some studies have revealed that lunasin acts as an integrin antagonist and affects downstream signaling pathways [21,58,67,94,98].

Nuclear factor kappa B (NF-κB) is a signaling pathway activated by integrins αvβ3 and α5β1 to regulate cell survival, along with angiogenesis- and inflammation-related gene expression [102,103]. Lunasin inhibits NF-κB signaling in colon cancer cell lines, an effect potentially associated with its binding to α5β1 integrin [98]. After lunasin treatment, Western blot assays showed a decrease in phosphorylated focal adhesion kinase (FAK), phosphorylated extracellular regulated kinase (ERK), and the p50 and p65 subunits of nuclear NF-κB, along with increased NF-κB inhibitor alpha (IκB-α) expression [98]. In THP-1 macrophages, lunasin inhibited Akt-mediated NF-κB activation via binding to αvβ3 integrin and inhibiting Akt and p65 activation [58]. Lunasin also inhibits NF-κB activation in breast cancer cells through increasing IκB expression, limiting its phosphorylation and degradation, while further reducing p65 nuclear translocation. In the same study, lunasin inhibited integrin-mediated FAK/Akt/ERK pathways by downregulating FAK, Src, ERK, and Akt [67].

Other studies support the lunasin inhibition of integrin-mediated FAK/Akt/ERK signaling pathways [21,94]. In the H661 lung cancer cell line, lunasin prevented β1 and β3 subunits from interacting with pFAK, integrin-linked protein kinase (ILK), and kindlin, while further blocking FAK, Akt, and ERK1/2 phosphorylation [94]. In A375 and B16-F10 melanoma cell lines, immunoblots identified a decrease in FAK, Akt, and ERK phosphorylation [21]. Cancer-initiating cells were more sensitive to phosphorylation being inhibited in these proteins, and lunasin interaction with the αv subunit suppressed the interaction between β1 and β3 subunits with pFAK and ILK [21]. Collectively, these data suggest that lunasin antagonizes integrin-mediated signaling pathways. Integrins are important to cancer progression and metastasis [20]. Therefore, integrin antagonists have attracted interest as a target for cancer treatment. Monoclonal antibodies and synthetic peptides have been developed as integrin antagonists; however, some clinical trials have failed to translate the antitumor effect [104]. Lunasin is a promising alternative integrin antagonist in cancer therapy.

### 4.2. Cell Cycle and Apoptosis Regulation

Responses related to cell cycle regulation under lunasin treatment have varied across studies and cell lines. Such differences are expected given the complexity of cancer pathophysiology and tumor heterogeneity. We also note that lunasin concentrations and sources vary between studies, probably influencing the different responses. Lunasin arrests the cell cycle at sub-G0/G1 [83], G1/S [68,86,87], and G2/M [46,66] phases in some studies, but does not affect cell cycle progression in others [21,84].

The regulation of cell-cycle-related gene expression has been reported for lunasin. The lunasin treatment of MDA-MB-231 cells increased retinoblastoma (Rb) transcriptional corepressor 1 (RB1) expression but decreased cyclin E1 (CCNE1), cyclin dependent kinase 2 (CDK2), cyclin dependent kinase 4 (CDK4), cell division cycle 25A (CDC25A), and E2F transcription factor 1 (E2F1) expression [86]. Increased RB1 expression was also observed in a later study by Hsieh et al. [87]. Cyclin D1 gene expression (CCND1) decreased in lunasin-treated HCT-116 cells [84]. Furthermore, lunasin has been observed to alter cyclin and CDKs expression. For example, cyclin D1, cyclin D3, CDK4, and cyclin-dependent kinase 6 (CDK6) expression all decreased in lunasin-treated MDA-MB-231 cells [13]. In H661 cells, lunasin treatment delayed cyclin D1 and CDK4 expression [68]. Lunasin also has differential influence on CDK inhibitor expression. Neither p21 nor p27 expression were affected in two studies [13,46]. However, another study noted a slight increase in p21, although p27 remained unaffected [83]. Increased p21 expression was similarly observed in NIH-3T3 cells transfected with E1A oncogene [16]. In contrast, Dia and de Mejia [66] recorded the increased expression of both p21 and p27 in KM12L4 cells, while McConnell et al. [68] reported increased p27 expression in H661 cells. Another effect of lunasin on the cell cycle is by activating the tumor suppressor Rb through inhibiting its phosphorylation [68,96]. Rb protein is responsible for the G1/S checkpoint and is inactivated via the phosphorylation by cyclin/CDK complexes [105]. Thus, lunasin arrests cancer cells in the G1/S phase through modulating cell-cycle gene expression. This outcome possibly stems from both epigenetic mechanisms and integrin antagonism, as integrin signaling pathways are related to the control of cell cycle progression [106].

Lunasin has distinct effects on apoptosis induction across cancer cell lines. In the H661 cell line, lunasin treatment did not induce apoptosis but did have antiproliferative effects [68]. Lunasin lowered A375 and SKMEL-28 cancer-initiating cell subpopulations but it did not induce apoptosis [77]. However, apoptosis has been induced in other cancer cell lines. In the L1210 leukemia cell line, lunasin-enriched soy flour-induced apoptosis, condensing chromatin, and increasing the expression of initiator caspases 8/9 and effector caspase-3 [46]. Lunasin also induced apoptosis in KM12L4 colon cancer cell lines, causing morphological alterations, such as nuclear condensation and DNA fragmentation [66]. Lunasin activated the mitochondrial pathway through upregulating caspase-3, pro-apoptotic Bax, and nuclear clusterin expression while downregulating anti-apoptotic Bcl-2 expression; cytosolic cytochrome c release was also elevated and caspases 9/3 were activated [66]. Lunasin-induced apoptosis and caspase-3 activation have also been reported in HCT-116 colon cancer cell lines [83]. Finally, lunasin successfully activated PTEN-mediated apoptosis in MCF-7 breast cancer cells [85] and HCT-116 cancer stem cells [84].

### 4.3. Antioxidant, Anti-Inflammatory, and Immunomodulatory Activities

The antioxidant activity of peptide lunasin has been associated with its amino acid composition. In the study by Hernández-Ledesma et al. [15], lunasin inhibited lipid peroxidation, exhibited 2,2′-azino-bis(3-ethylbenzothiazoline-6-sulfonic acid) radical cation (ABTS•+) scavenging activity, and blocked the lipopolysaccharide-induced generation of reactive oxygen species (ROS). Lunasin antioxidant activity was further demonstrated when it successfully scavenged peroxyl and superoxide radicals, chelated ferrous ions, and lowered intracellular ROS levels [56]. Fernández-Tomé et al. [57] reported that lunasin decreased both intracellular ROS and protein carbonyl levels, increased cytosolic glutathione levels, and prevented an elevation of glutathione peroxidase and catalase activity under oxidative stress.

Lunasin exerted anti-inflammatory properties through decreasing nitric oxide (NO) and prostaglandin E2 (PGE2) production, as well as through the expression of inducible NO synthase (iNOS) and cyclooxygenase-2 (COX-2), in macrophage cell lines RAW 264.7 [70,107] and THP-1 [58]. Lunasin also inhibited the production of pro-inflammatory cytokines interleukin-6 and interleukin-1β, and its anti-inflammatory properties were associated with suppressing the NF-κB pathway [70]. The relationship between lunasin anti-inflammatory activity and this pathway was further demonstrated in a study showing that its interaction with αvβ3 integrin downregulated Akt-mediated NF-κB activation [58]. Therefore, the anti-inflammatory properties of lunasin are likely related to its integrin antagonism. Importantly, studies with lunasin fragments suggest that the entire sequence is necessary for anti-inflammatory effects [15]. Likewise, synthetic lunasin did not alter NO production [15], whereas purified lunasin did [58,70,107]. Several studies have also reported that lunasin can exert anti-inflammatory activity through downregulating tumor necrosis factor alpha (TNF-α) production [15,58].

Lunasin exhibits immunomodulatory activity against cancer, specifically through combining with cytokines interleukin-2 (IL-2) and interleukin-12 (IL-12); this combination has a synergistic effect that stimulates natural killer (NK) cells to enhance interferon gamma (IFNγ) production [18]. The lunasin/IL-2/IL-12 combination also increased the expression of granzyme B (GZMB) and the granulocyte–macrophage colony stimulating factor (CSF2), while decreasing transforming growth factor beta 1 (TGFB1) and transforming growth factor beta receptor 2 (TGFBR2) expression [18]. Thus, the immunomodulatory effect of lunasin is linked to its modulation of gene expression. Lunasin combined with IL-12 also elevated H3 acetylation in the interferon gamma (IFNG) locus and lowered H3 acetylation in the TGFB1 locus, indicating an epigenetic mechanism [18]. However, the RGD motif and aspartic acid (D)-tail were not associated with a synergistic enhancement of IFNγ secretion, because a peptide lacking both regions had a comparable effect as full-length lunasin. Therefore, the immunomodulatory activity of lunasin is associated with its N-terminus and/or central regions. The lunasin mechanism of action against cancer is summarized in Figure 3.

## 5. Lunasin Products and Potential Combination Therapeutics

No commercial products are currently available that exploit the therapeutic application of lunasin against cancer. Commercially available lunasin products do not focus on anticancer activity but are instead sold as dietary supplements that emphasize antioxidant, anti-inflammatory, and cholesterol-lowering activities. Such products include LunaCell^®^ (Simplesa Nutrition Corp., Miami, FL, USA), LunaRich X™ (Reliv International, Inc., Chesterfield, MO, USA), and Lunasin Cellular Health Formula (Carefast Products, Inc., Las Vegas, NV, USA).

There is a potential to develop lunasin nutraceutical products with chemopreventive and therapeutic actions against cancer. Other natural components of soybean that exhibit anticancer activity can be combined to achieve an improved effect. Epidemiological studies suggest that Asian countries consuming soy-based foods, especially isoflavones, have lower incidence of some types of cancer [108]. Genistein is one of the major soy isoflavones with promising chemopreventive and cancer therapeutic applications [109,110]. Although there are questions about potential harmful effects, some clinical studies indicate that genistein is safe at the dose needed to be bioactive [111,112,113]. Such as lunasin, the anticancer mechanism of action of genistein involves cell cycle regulation, the induction of apoptosis, the modulation of signal transduction pathways and anti-inflammatory effect [108,114]. The effects of lunasin and genistein were evaluated using the HCT-116 colon cancer cell line [84]. They demonstrated a decrease in cell viability, apoptosis induction, a decrease in colonosphere formation, and upregulate PTEN mRNA levels [84]. Differently, while genistein induced G1/S arrest [84], lunasin did not affect cell cycle progression. Similarities of lunasin and genistein activities has been investigated in malignant and non-malignant mammary cell lines [85]. Both induced PTEN-mediated apoptosis and promoted E-cadherin and β-catenin non-nuclear localization, but lunasin did not promote p53 nuclear localization or inhibit Wnt1-induced cellular proliferation [85]. Thus, combining the different mechanisms of action of lunasin and genistein can be interesting regarding anticancer activity. The combined effects of lunasin and genistein can also be investigated in terms of diet. The combination of specific diets and bioactive compounds has been discussed to improve health benefits [115]. Investigating the lunasin chemopreventative and therapeutic effects together with the consumption of diets rich in isoflavones is an interesting alternative.

## 6. Conclusions

The anticancer properties of lunasin have been studied for over 20 years. Increasing evidence suggests that lunasin is a promising candidate for the prevention and treatment of multiple cancers. Lunasin has multiple notable attributes, including epigenetic effects, metastasis inhibition, and action against cancer stem cells. However, the applications of lunasin in cancer therapy still encounter limitations, necessitating the improved understanding of its structure and activity. On a structural level, the entire lunasin sequence is important for anticancer activity. The most important PTM in the lunasin structure is the disulfide bond between C_10_ and C_22_ residues, observed across lunasin obtained from various sources. The thermostability of lunasin is also a valuable characteristic for simplifying processing and storage conditions. Lunasin is intrinsically disordered, a trait that favor structural plasticity and multiple actions. However, lunasin may vary in structural content depending on its origin, presenting a challenge for characterization, and likely generating differences in activity.

Similar to other therapeutic polypeptides, lunasin application faces hurtles related to pharmacokinetics, mainly regarding oral bioavailability. The issues stem from its high susceptibility to degradation by digestive enzymes. However, lunasin has good absorption ability that allow for other routes besides oral administration. Lunasin also exhibits strong distribution capacity, able to reach numerous tissues and cross the blood–brain barrier. The latter requires special attention during drug development, because lunasin can act on the central nervous system with potentially adverse events. Although some studies suggest that lunasin is safe, its safety remains relatively inconclusive. Therefore, further studies are required to evaluate the safety of its therapeutic application. In terms of its activity, lunasin exerts distinct effects depending on cancer types, clearly suggesting a complex mechanism of action and involvement of different pathways. We currently know of two major molecular partners in lunasin anticancer activity: histones H3 and H4 are involved in the epigenetic mechanism, whereas integrin antagonism is associated with inhibition of integrin-mediated signaling pathways. Nevertheless, available lunasin products are not intended for cancer applications. We conclude that although more research is necessary to clarity lunasin anticancer action and safety as a therapeutic option, it is a promising candidate for developing novel cancer treatments. Especially, due to its selectivity, lunasin may also be promising in cancer therapies combined with other drugs.

## Figures and Tables

**Figure 1 ijms-23-09548-f001:**
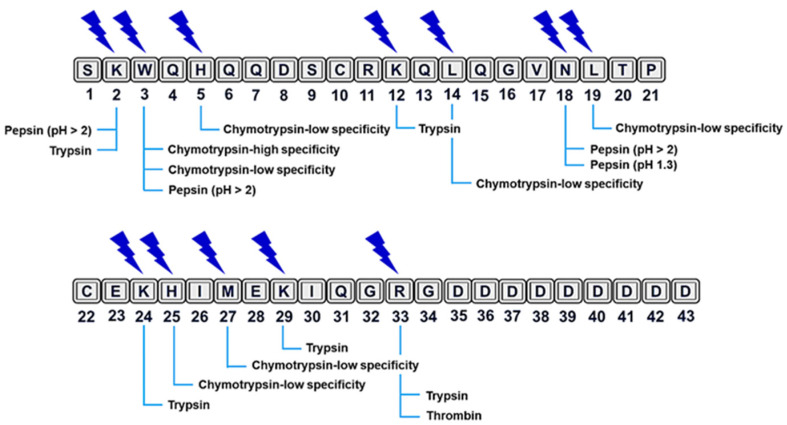
Prediction of cleavage sites for digestive and plasmatic proteases in the lunasin sequence. PeptideCutter (available from: https://web.expasy.org/peptide_cutter/, accessed on 8 June 2021) was utilized for prediction. Predicted cleavage sites are indicated with blue lightning bolts, and putative enzymes for each site are indicated along the bottom of the figure.

**Figure 2 ijms-23-09548-f002:**
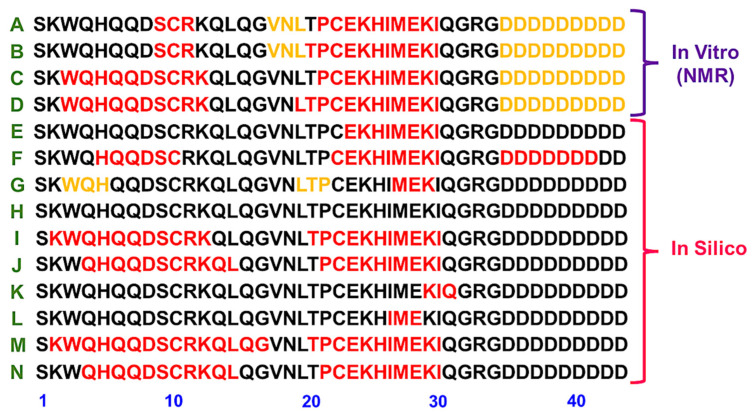
Lunasin secondary structure content location from the published literature. (**A**,**B**) Secondary structure elements identified by NMR for the recombinant lunasin at pH 3.5 without or with disulfide bond, respectively [36]. (**C**,**D**) Secondary structure elements identified by NMR for the recombinant lunasin at pH 6.5 without or with disulfide bond, respectively [36]. (**E**) Proposed α-helix motif with similarity to chromatin-binding proteins [14]; (**F**) Structural content observed by molecular dynamics simulations [37]; (**G**,**H**) Reduced and oxidized forms of the extended lunasin model analyzed by molecular dynamic studies in water [24], respectively. (**I**,**J**) Reduced and oxidized forms from the predicted lunasin model analyzed by molecular dynamic studies in water [24], respectively. (**K**,**L**) Reduced and oxidized forms, respectively, from the extended lunasin model analyzed by molecular dynamic studies in mixture of water and TFE [24]. (**M**,**N**) Reduced and oxidized forms, respectively, from the predicted lunasin model analyzed by molecular dynamic studies in mixture of water and TFE [24]. Residues in red are α-helix motifs and residues in yellow are from a β-strand.

**Figure 3 ijms-23-09548-f003:**
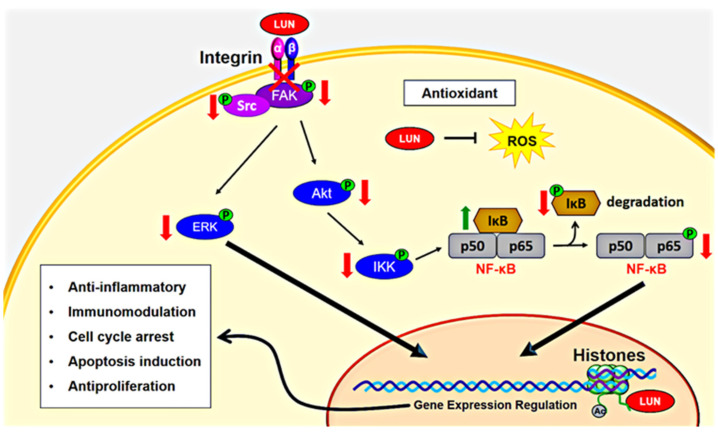
Simplified lunasin mechanism of action against cancer. Lunasin (LUN) has antioxidant capacity, lowering intracellular reactive oxygen species (ROS) levels. Furthermore, lunasin has epigenetic effects, binding to the N-terminus of deacetylated histones H3 and H4, as well as regulating acetylation (Ac) and deacetylation of specific lysines. Lunasin also binds to and antagonizes integrin-mediated signaling pathways, such as Ras/MEK/ERK and PI3K/Akt. Lunasin antagonism on integrins decreases FAK, Src, ERK, and Akt phosphorylation. Integrin antagonism also allows lunasin to inhibit the NF-κB pathway by limiting IκB kinase (IKK) activation. The IKK complex phosphorylates and promotes IκB degradation, releasing p50 and p65 subunits of NF-κB for nuclear translocation and the activation of gene expression. Thus, the epigenetic and integrin-antagonistic actions of lunasin regulate the expression of genes involved in its anticancer properties, including anti-inflammation, immunomodulation, cell cycle arrest, apoptosis induction, and antiproliferation.

**Table 1 ijms-23-09548-t001:** Structure–activity relationship of lunasin. Anticancer-related activities reported are correlated with all parts of its sequence.

Part of the Sequence	Activity	Ref.
**SKWQHQQDSCRKQLQGVNLTPCEKHIME-KIQGRGDDDDDDDDD**	Cytotoxicity	[13]
	Anti-transformation	[14,16]
	Anti-inflammatory	[15]
**SKWQHQQDSCRKQLQGVNLTPCEKHIMEKIQGDDDDDDDDD**	Inhibition of histone H4 acetylation	[14]
**SKWQHQQDSCRKQLQGVNLTPCEKHIMEKIQG**	Antioxidant	[15,17]
	Immunomodulatory	[18]
** EKHIMEKIQG **	Chromatin binding	[14]
** RGD **	Cell internalization	[14]
** DDDDDDDDD **	Inhibition of histone H3 acetylation	[5,13]

Portions of the lunasin sequence are represented in different colors. N-terminus, central portion, RGD motif, and aspartic acid (D)-rich C-terminus are represented with residues in black, red, blue, and green, respectively.

**Table 2 ijms-23-09548-t002:** Lunasin activity against normal and immortalized cells.

Cell Line	Lunasin Source	Concentration	Assay	Effects	Ref.
NIH/3T3	Synthetic	10 µM	Coulter counter	No effect	[16]
	Synthetic	100 nM	MTT	No effect	[61]
	Synthetic	0.01–10 µM	MTT	No effect	[62]
SRA 01/04	Recombinant	1–100 µM	MTT	No effect	[63]
	Recombinant	20, 40, 80 µM	Apoptosis detection (Annexin V-PE/7-AAD)	No effect	[64]
CCD-33Co	Soybean (~90%)	1–100 µM	MTS	No effect	[78]
	Soybean (~90%)	0–100 µM	MTS	No effect	[66]
BEAS-2B	Soybean (>99%)	1–100 µM	MTS	No effect	[68]
HBE135-E6E7	Soybean (>99%)	1–100 µM	MTS	No effect	[68]
MCF-10A	Synthetic	0–320 µM	MTT	No effect	[67]
EA.hy926	Recombinant	0–120 µM	MTT	No effect	[65]
HEK-293	Synthetic	0.1–100 µM	MTT	↓ Proliferation at ˃100 µM	[74]
Chondrocyte	Synthetic	50–500 µM	MTT	↓ Proliferation at ˃500 µM	[72]
3T3-L1	Synthetic	0.1–25 µM	MTT	No effect	[69]
Synovial Fibroblast	Synthetic	0–200 µM	Crystal violet staining and apoptosis detection (annexin V-FITC/PI)	↓ Proliferation (IC_50_ 153.3 µM). Apoptosis induction	[73]
RAW 264.7	Synthetic	0.2–200 µM	MTT	No effect	[15]
	Soybean (85.3%)	10–50 µM	MTS	No effect	[70]
	Synthetic	0.2–200 µM	MTT	No effect	[71]
	Synthetic	10–100 µM	MTT	No effect	[17]

The downward arrow (↓) indicates decrease. Abbreviations: 3T3-L1, adipocyte differentiated mouse embryonic fibroblast; 7-AAD, 7-amino-actinomycin D; BEAS-2B, human bronchial epithelial cells; CCD-33Co, human colon fibroblast; EA.hy926, human umbilical vein cell line; FITC, fluorescein isothiocyante; IC_50_, half-maximal inhibitory concentration; HBE135-E6E7, human bronchial epithelial cells; HEK-293, human embryonic kidney cells; MCF-10A, human breast epithelial cells; MTS, 3-(4,5-dimethylthiazol-2-yl)-5-(3-carboxymethoxyphenyl)-2-(4-sulfophenyl)-2H-tetrazolium; MTT, 3-(4,5-dimethylthiazol-2-yl)-2,5-diphenyltetrazolium bromide; NIH-3T3, mouse embryonic fibroblast; PE, phycoerythrin; PI, propidium iodide; RAW 264.7, mouse monocyte/macrophage-like cell line; Ref., reference; SRA 01/04, human lens epithelial cell line.

**Table 3 ijms-23-09548-t003:** Lunasin activity against cancer cell lines.

Tissue	Cell Line	Lunasin Source	Concentration	Assay	Effects	Ref.
Colon	HT-29 (Human)	Soybean (~90%)	1–100 µM	MTS and crystal violet staining	↓ Proliferation (IC_50_ 61.7 µM) and morphologic alterations	[78]
		Soybean (>90%)	1–100 µM	MTS	↓ Proliferation (IC_50_ 61.7 µM)	[66]
		Synthetic	10–200 µM	MTT	↓ Proliferation	[51]
	HCT-116 (Human)	Soybean (>90%)	1–100 µM	MTS	↓ Proliferation (IC_50_ 26.3 µM)	[66]
		Recombinant	1–100 µM	MTT	↓ Proliferation (IC_50_ 64.25 μM)	[63]
		Synthetic	2 µM	Trypan blue exclusion, apoptosis detection (annexin V) and colonosphere formation	↓ Proliferation. ↑ Apoptotic cells. ↓ Colonosphere formation	[84]
		Recombinant	20, 40 and 80 µM	Apoptosis detection (annexin V-PE/7-AAD)	↑ Apoptotic cells	[64]
		Synthetic	5–160 µM	MTT, tumorsphere formation and apoptosis detection (annexin V/PI)	↓ Proliferation (IC_50_ 107.5 µM). ↓ Tumorsphere formation (IC_50_ 161 µM). ↑ Apoptotic cells	[83]
	HCT-116OxR (Human)	Soybean (>90%)	1–100 µM	MTS	↓ Proliferation (IC_50_ 31.6 µM)	[66]
	KM12L4 (Human)	Soybean (>90%)	1–100 µM	MTS	↓ Proliferation (IC_50_ 13 µM)	[66]
	KM12L4OxR (Human)	Soybean (>90%)	1–100 µM	MTS	↓ Proliferation (IC_50_ 34.7 µM)	[66]
	RKO (Human**)**	Soybean (>90%)	1–100 µM	MTS	↓ Proliferation (IC_50_ 21.6 µM)	[66]
	RKOOxR (Human**)**	Soybean (>90%)	1–100 µM	MTS	↓ Proliferation (IC_50_ 38.9 µM)	[66]
	Caco-2 (Human)	Synthetic	0.5–25 µM	MTT	No effect	[56]
		Synthetic	10–200 µM	MTT	Modest decrease in viability (24 h and 48 h). No cytotoxicity (72 h)	[51]
Stomach	AGS (Human)	Synthetic	10–200 µM	MTT	Modest decrease in viability	[51]
Liver	HepG2 (Human)	Synthetic	0.5–50 µM	Crystal violet staining	No effect	[57]
		Recombinant	1–100 µM	MTT	No effect	[63]
		Recombinant	20, 40 and 80 µM	Apoptosis detection (annexin V-PE/7-AAD)	Negligible early apoptosis induction	[64]
Breast	MCF-7 (Human)	Synthetic	10 µM	Coulter counter	No effect	[16]
		Synthetic	2 µM	TUNEL and mammosphere formation	↑ Apoptotic cells. No influence on mammosphere formation	[85]
		Synthetic	0–320 µM	MTT	↓ Proliferation (IC_50_ 431.9 µM)	[67]
	MDA-MB-231 (Human)	Synthetic	0.1–200 µM	MTT and apoptosis detection (annexin V/7-AAD)	↓ Proliferation (IC_50_ 181 µM). No apoptosis induction	[86]
		Synthetic	1–200 µM	MTT and apoptosis detection (annexin V/7-AAD)	↓ Proliferation (IC_50_ 181 µM). No apoptosis induction	[87]
		Synthetic	10–200 µM	MTT	↓ Proliferation (IC_50_ 181 µM)	[13]
		Recombinant	1–100 µM	MTT	↓ Proliferation (IC_50_ 56.73 μM)	[63]
		Synthetic	0–320 µM	MTT	↓ Proliferation (IC_50_ 194.9 µM)	[67]
		Recombinant	20, 40 and 80 µM	Apoptosis detection (annexin V-PE/7-AAD)	↑ Apoptotic cells	[64]
	4T1 (Mouse)	Synthetic	1–50 µM	MTT	No effect	[69]
Skin (Melanoma)	A375 (Human)	Soybean (>99%)	100 µM	MTS and colony Formation	No effect on cell proliferation. ↓ Colony formation (37%)	[77]
	SKMEL-28 (Human)	Soybean (>99%)	100 µM	MTS and colony formation	No effect on cell proliferation. ↓ Colony formation (23%)	[77]
	B16-F10 (Mouse)	Soybean	100 µM	Oncosphere formation and Transwell invasion	↓ Oncosphere formation (29%). ↓ Invasion (60%)	[21]
	B16-F0 (Mouse)	Soybean (>99%)	1–100 µM	MTS and colony formation	Modest decrease in viability. ↓ Colony formation	[88]
Lung	NCI-H661 (Human)	Soybean (>99%)	1–100 µM	MTS, colony formation and apoptosis detection (annexin V-Cy3™ and 6-CFDA)	↓ Proliferation (IC_50_ 63.9 µM) ↓ Number of colonies	[68]
	NCI-H1299 (Human)	Soybean (>99%)	1–100 µM	MTS and colony formation	No cytotoxicity. ↓ Number and size of colonies.	[68]
	NCI-H460 (Human)	Soybean (>99%)	1–100 µM	MTS and colony formation	No cytotoxicity. ↓ Colony size.	[68]
	A549 (Human)	Soybean (>99%)	1–100 µM	MTS and colony formation	No cytotoxicity. ↓ Colony number	[68]
	LLC (Mouse)	Soybean (>99%)	1–100 µM	MTS and colony formation	Modest decrease in viability. ↓ Colony formation.	[88]
Blood (Leukemia)	THP-1 (Human)	Soybean (>95%)	10–100 µM	MTS	No effect	[59]
	L1210 (Mouse)	Soybean (~98%)	0–100 µM	CCK-8	↓ Proliferation (IC_50_ 14 µM)	[46]

The downward (↓) and upward (↑) arrows indicate decrease and increase, respectively. Abbreviations: 6-CFDA, 6-carboxyfluorescein diacetate; 7-AAD, 7-amino-actinomycin D; CCK-8, cell counting kit-8; IC_50_, half-maximal inhibitory concentration; MTS, 3-(4,5-dimethylthiazol-2-yl)-5-(3-carboxymethoxyphenyl)-2-(4-sulfophenyl)-2H-tetrazolium; MTT, 3-(4,5-dimethylthiazol-2-yl)-2,5-diphenyltetrazolium bromide; PE, phycoerythrin; PI, propidium iodide; PTEN, phosphatase and tensin homolog gene; Ref., reference; TUNEL, terminal deoxynucleotidyl transferase dUTP nick end labeling.

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
