# Peer review of "Lunasin as a Promising Plant-Derived Peptide for Cancer Therapy"

_ijms, 2022, doi:10.3390/ijms23179548_

Round 1
Reviewer 1 Report
In the present review “Lunasin as a promising plant-derived peptide for cancer therapy”, Stephanny Miranda Alves de Souza and colleagues evaluated the available research on lunasin’s structure and mechanism of action, which should be useful for the development of lunasin-based therapeutic products. They also provided an overview of research on lunasin pharmacokinetics and safety. Finally, the authors briefly discussed commercially available lunasin products and patents.
Overall, I think that the manuscript is intriguing, well-written (within the scope of this journal), well-structured and the data are of potential clinical relevance. I would like to congratulate the authors on their work.
I have a little question/curiosity.
“Oriental diet” is particularly abundant in soy isoflavones. So, the potential nutraceutical protective effects of lunasin against cancer and NCDs could be analyzed, from a translational point of view, in combination with soy isoflavones (i.e. genistein). Please discuss this aspect in the paper considering for your convenience these references (Marini, H. et al. J. Clin. Endocrinol. Metab. 2008, 93, 4787-4796; Rasheed, S. et al. J Food Biochem. 2022, May 17: e14228; G. Mohan Shankar et al. Front Pharmacol. 2022, 12:809308; Marini, H.R. Nutrients, 2022, 14, 1550).
Author Response
In the present review “Lunasin as a promising plant-derived peptide for cancer therapy”, Stephanny Miranda Alves de Souza and colleagues evaluated the available research on lunasin’s structure and mechanism of action, which should be useful for the development of lunasin-based therapeutic products. They also provided an overview of research on lunasin pharmacokinetics and safety. Finally, the authors briefly discussed commercially available lunasin products and patents. Overall, I think that the manuscript is intriguing, well-written (within the scope of this journal), well-structured and the data are of potential clinical relevance. I would like to congratulate the authors on their work.
I have a little question/curiosity. “Oriental diet” is particularly abundant in soy isoflavones. So, the potential nutraceutical protective effects of lunasin against cancer and NCDs could be analyzed, from a translational point of view, in combination with soy isoflavones (i.e. genistein). Please discuss this aspect in the paper considering for your convenience these references (Marini, H. et al. J. Clin. Endocrinol. Metab. 2008, 93, 4787-4796; Rasheed, S. et al. J Food Biochem. 2022, May 17: e14228; G. Mohan Shankar et al. Front Pharmacol. 2022, 12:809308; Marini, H.R. Nutrients, 2022, 14, 1550).
Answer: Thanks to reviewer 1 for the comments and for raising an interesting discussion. We believe that the changes made in response to your question contributed substantially to the article.
To address your question, the title of section 5 has been changed to “5. Lunasin Products and Potential Therapeutic Combinations” and the following new text has been added: “There is a potential to develop lunasin nutraceutical products with chemopreventive and therapeutic actions against cancer. Other natural components of soybean that exhibit anticancer activity can be combined to achieve an improved effect. Epidemiological studies suggest that Asian countries consuming soy-based foods, especially isoflavones, have lower incidence of some types of cancer [108]. Genistein is one of the major soy isoflavones with promising chemopreventive and cancer therapeutic applications [109, 110]. Although there are questions about potential harmful effects, some clinical studies indicate that genistein is safe at the dose needed to be bioactive [111-113]. Such as lunasin, the anticancer mechanism of action of genistein involves cell cycle regulation, induction of apoptosis, modulation of signal transduction pathways and anti-inflammatory effect [108, 114]. Effects of lunasin and genistein were evaluated using the HCT-116 colon cancer cell line [84]. They demonstrated a decrease in cell viability, apoptosis induction, decrease in colonosphere formation and upregulate PTEN mRNA levels [84]. Differently, while genistein induced G1/S arrest [84], lunasin did not affect cell cycle progression. Similarities of lunasin and genistein activities has been investigated in malignant and non-malignant mammary cell lines [85]. Both induced PTEN-mediated apoptosis and promoted E-cadherin and β-catenin non-nuclear localization, but lunasin did not promote p53 nuclear localization or inhibited Wnt1-induced cellular proliferation [85]. Thus, combining different mechanisms of action of lunasin and genistein can be interesting for the anticancer activity. Combined effects of lunasin and genistein can also be investigated in terms of diet. Combination of specific diets and bioactive compounds has been discussed to improve health benefits [115]. Investigate the lunasin chemopreventive and therapeutic effects together with the consumption of diets rich in isoflavones is an interesting alternative”.
This new text has been highlighted with red color. Additionally, from this text, new references were included in the manuscript, which include those indicated by the reviewer and related ones.
Reviewer 2 Report
No doubt, cancerous diseases are one of the leading health problems. Therefore the manuscript could have general interest. However several parts of the paper are wordy and uninteresting. The specific remarks:
Lunasin structure: "Lunasin cannot be considered a peptide" is false. Lunasin is a polypeptide. Fig. 1. is a little bit confusing. The side-chain modification mentioned in line 154 mostly are unnatural ones, mixed with the real PTMs. The formation of aberrant disulfide bridges during the storage of a protein is unusual.
Secondary structure:
There are significant differences in the structures summarized in Fig. 3. Probably the NMR studies describe almost exactly the real structure.
Pharmacokinetics:
There are numerous alternatives to oral administration.
Mechanisms:
Table 1. is exhaustive. Table 2. too. According to my opinion, it should be simplified.
line 408 NH2 should be subscript.
Table 3. Setting out the patents is unnecessary.
Author Response
No doubt, cancerous diseases are one of the leading health problems. Therefore, the manuscript could have general interest. However, several parts of the paper are wordy and uninteresting. The specific remarks:
- Lunasin structure: "Lunasin cannot be considered a peptide" is false. Lunasin is a polypeptide.
Answer: We appreciate the reviewer’s observation. To address this issue, we chose to remove the sentence “Lunasin cannot be considered a peptide according to the definition of the US Food and Drug Administration because it contains more than 40 amino acids in its polypeptide chain” from the manuscript, as it is not essential information, and could be misunderstood. The sentence had initially been included to raise an FDA peptide definition issue (https://www.fda.gov/drugs/regulatory-science-action/impact-story-developing-tools-evaluate-complex-drug-products -peptides).
- Fig. 1. is a little bit confusing.
Answer: We appreciate the reviewer's observation and agree that the information in Fig. 1 could be better specified to avoid confusion. To answer the question, we chose to replace the figure with a table, as we believe that the information would make it clearer.
Thus, a new "Table 1. Structure-activity relationship of lunasin. Anticancer-related activities reported are correlated with all parts of its sequence." has been added, as well as the related text has been adjusted, which is highlighted in red.
- The side-chain modification mentioned in line 154 mostly are unnatural ones, mixed with the real PTMs.
Answer: We appreciate the reviewer's question and agree that the information could be better specified. To address this issue, we have included new sentences to clarify the points raised, which are highlighted in red in the text.
- The formation of aberrant disulfide bridges during the storage of a protein is unusual.
Answer: We appreciate and agree the reviewer's comment. To address this question, we chose to remove “and storage” from the sentence.
- Secondary structure: There are significant differences in the structures summarized in Fig. 3. Probably the NMR studies describe almost exactly the real structure.
Answer: We appreciate the reviewer's comment and agree that the structural information could be better described. To answer the question, we rewrote the information and reorganized the figure to highlight the results obtained from NMR studies. Since a considerable number of in silico studies are available, they were also considered.
- Pharmacokinetics: There are numerous alternatives to oral administration.
Answer: We appreciate the reviewer's comment. Since we understand the importance of discussing possible alternatives to oral administration in the text, to answer this question, we have added the following sentences and references “In addition to the co-administration with protease inhibitors, other strategies can be employed to make feasible oral administration of lunasin. Some strategies to enhance oral bioavailability and increase resistance to proteolytic digestion may include chemical modifications (e.g., D-amino acids), mucoadhesive systems (e.g., chitosan, carbopol, cellulose derivatives) and carrier systems (e.g., liposomes, hydrogels, polymeric micro- and nanoparticles) [49, 50].”
- Mechanisms:
Table 1. is exhaustive. Table 2. too. According to my opinion, it should be simplified.
Answer: We appreciate the reviewer's comment. To address this question, both tables have been simplified as can be observed in the new version of the manuscript.
- Line 408 NH2 should be subscript.
Answer: We appreciate the observation. At line 408, NH2 was changed to subscript (NH2) as requested.
- Table 3. Setting out the patents is unnecessary.
Answer: The part of the text and the table related to patents were removed from the manuscript.
The title of section 5 has been changed from “5. Lunasin Products and Patents” to "5. Lunasin Products and Potential Combination Therapeutics", as well as new text was written in this section to address reviewer 1's question.
Round 2
Reviewer 2 Report
The authors have addressed all the points asked for. So, this time the paper looks good for publication.